# Using Einkorn and Tritordeum Brewers’ Spent Grain to Increase the Nutritional Potential of Durum Wheat Pasta

**DOI:** 10.3390/foods10030502

**Published:** 2021-02-26

**Authors:** Francesca Nocente, Chiara Natale, Elena Galassi, Federica Taddei, Laura Gazza

**Affiliations:** CREA Research Centre for Engineering and Agro-Food Processing, Via Manziana, 30 00189 Rome, Italy; francesca.nocente@crea.gov.it (F.N.); chiara.natale@tiscali.it (C.N.); elena.galassi@crea.gov.it (E.G.); federica.taddei@crea.gov.it (F.T.)

**Keywords:** brewers’ spent grain, dietary fiber, einkorn, functional pasta, tritordeum, upcycling

## Abstract

Brewers’ spent grain (BSG), the major by-product of the brewing industry, can be used as a functional ingredient to increase the nutritional value of cereal-based products. In this work, micronized BSG from the einkorn and tritordeum brewing processes were characterized and used to produce four macaroni pasta formulations enriched with BSG at ratios of 5 g and 10 g/100 g of semolina. Einkorn BSG showed the highest values for all the parameters analyzed—proteins, total dietary fiber (TDF) and total antioxidant capacity (TAC)—except for β-glucan. TDF increased up to 42 and 68% in pasta samples enriched with 10% of BSG from tritordeum and einkorn, respectively. The replacement of 10% of semolina with BSG from both cereals significantly increased the β-glucan content and TAC values. Finally, the addition of BSG from einkorn and tritordeum affected to a minimal extent the sensory properties of cooked pasta, which showed higher values of optimal cooking time and cooking loss, but lower total organic matter compared to semolina pasta. Results from the sensorial judgment fell in the good quality ranges for durum wheat pasta; the incorporation of 10% of einkorn BSG resulted in the best compromise in terms of technological, nutritional and sensorial aspects of enriched pasta.

## 1. Introduction

The agri-food sector presents a major opportunity for the development of a circular economy, since waste and by-products, still rich in carbohydrates, proteins, lipids and complex nutraceuticals, can be managed to realize new valuable products, increasing the sustainability of the food and non-food chains [1,2,3]. According to the latest data provided by Eurostat, Italy is the leader in the use of recycled materials, with the highest share of circular materials used by the manufacturing system—the average percentage of waste recycling is around 77% of the total amount of waste [4]. Brewers’ spent grain (BSG) is the major by-product generated from the beer brewing process and it is estimated that almost 3.4 million tons of BSG are produced annually in the European Union, of which 288,000 tons/year are used in Italy [5,6]. BSG consists of insoluble grain components, mainly cereal seed coat, pericarp and husk, obtained after the extraction of worth. BSG is rich in fiber (30–50% *w*/*w*) and protein (19–30% *w*/*w*), but essential amino acids (e.g., methionine, phenylalanine, tryptophan, histidine and lysine), minerals (e.g., calcium, iron, magnesium, manganese, phosphorus, potassium, sodium), vitamins (e.g., biotin, choline, folic acid, niacin, pantothenic acid and riboflavin) and antioxidant compounds (tocols amd phenolic acids) are also present [5,7]. Variation in its chemical composition can depend on cereal variety or species, location and harvest time, malting and mashing conditions, drying method and the types of adjuncts used during the brewing process [8,9]. To date, the main use of BSG has been as fertilizer or animal feed. However, due to their valuable chemical composition, novel applications of BSG in different areas are expanding, such as extractions of proteins and fiber, sugars and bioactive molecules; energy production; and microorganism cultivation [5,10,11]. The increasing awareness of the relationship between the consumption of healthy foods and well-being led recently to exploiting the nutritional and functional potential of BSG in human nutrition. The use of barley BSG as adjunct in cereal-based food stuffs such as bread, pasta, cookies and snacks can increase their nutritional value, delivering healthier products towards a more sustainable food system [12,13,14,15,16]. Indeed, the incorporation into the human diet of functional compounds present in BSG such as arabinoxylans, β-glucan, tocols and phenolic acids, provides several benefits by contributing to lowering the risk of some diseases, including cancer, gastrointestinal disorders, diabetes, obesity and coronary heart disease [17,18,19,20]. Moreover, in recent years, the issue in agricultural biodiversity of reaching low-impact and sustainable agriculture has led to the rediscovery of several minor crops, such as einkorn and tritordeum, which also result in suitable raw material for malting and brewing [21,22]. Einkorn (*Triticum monococcum* L.) is an ancient diploid hulled wheat cultivated until the Bronze Age. In addition to its hardiness, it shows peculiar nutritional features, i.e., high protein content (16–18%), low nitrogen fertilization, high antioxidant content (carotenoids, tocols, conjugated polyphenols, alkylresorcinols and phytosterols), high contents of zinc and iron and a more digestible gluten with respect to the most cultivated wheats [23,24,25,26].

Tritordeum (x *Tritordeum* Ascherson and Graebner) is an amphiploid produced by crossing wild barley (*Hordeum chilense* Roem. and Schult) with either tetraploid (*Triticum turgidum* L. ssp. *durum* Desf.) or hexaploid wheat (*Triticum aestivum* L.). These hybridizations produce hexaploid and octoploid tritordeum, respectively [27]. Tritordeum could be a good novel raw material for the production of health-promoting foods, since, especially in the kernel’s outmost layers, it is characterized by a high content of fiber (particularly fructans, arabinoxylans and β-glucans to a minor extent), which ranges from 6 to 8%; moreover, it is a source of oleic acid (3–4%) and antioxidant compounds such as phenolic acids and xanthophylls, mainly lutein (4–6 µg/g) esterified with fatty acids, which improve its stability in storage and at high temperatures [28]. In addition, it has been shown to have far fewer gluten immunogenic peptides in comparison with wheat [29].

In this work, BSG recovered from the einkorn and tritordeum brewing processes were investigated as new functional ingredients with which to produce novel formulations of dry pasta with enhanced nutritional potential, which should able to tempt the international pasta market and meet the request of the upcycling of food industry’s waste material. Cooking properties, and textural, sensorial and nutritional characteristics of BSG-enriched durum wheat pasta, macaroni shape, were explored, to pursue the dual purposes of improving the sustainability of the brewing sector by valorizing its main process waste, and of producing pasta with enhanced nutritional value.

## 2. Materials and Methods

### 2.1. Raw Material

For this study, two different types of BSG were used to enrich pasta. One pilsner-type BSG (BSGE) was derived from a homemade brewing process, performed in a lab-scale plant, of a mix of two malts obtained from a hulled and a dehulled cultivar of einkorn, Norberto and Hammurabi, respectively. The other pilsner-type BSG (BSGT) was derived from the brewing of a malt from a hexaploid tritordeum line.

After the mashing step, BSG were stored at −20 °C and dried at 60 °C for 72 h. The BSG samples kilned to 7% of moisture were micronized at ≤700 μm sieve bya Pulverisette mill (Fritsch, Idar-Oberstein, Germany). Semolina was obtained from a mix of commercial durum wheat varieties grown in Italy during 2019 using the Buhler MLU 202 (Uzwil, Switzerland) plant.

### 2.2. Pasta-Making Process

Five pasta formulations were produced: S = semolina 100% used as reference; BSGE5 = semolina:BSGE 95:5 (*w*:*w*); BSGE10 = semolina:BSGE 90:10 (*w*:*w*); BSGT5 = semolina:BSGT 95:5 (*w*:*w*); BSGT10 = semolina: BSGT 90:10 (*w*:*w*) (Figure 1). To obtain proper consistency of the doughs for extrusion, semolina and semolina-BSG formulations were hydrated at different levels depending on the BSG origin. Specifically, the moisture of the dough of semolina pasta and of pasta enriched with BSG from einkorn was 42%, whereas the dough enriched with BSG from tritordeum was 38% water.

Pasta was obtained using a pilot plant consisting of an extruder (NAMAD, Rome, Italy) with a capacity up to 20 kg/h, equipped with a screw (45 cm in length, 4.5 cm in diameter), which ended with a Teflon-coated die (150 mm diameter) to produce macaroni shape, and of a dryer (AFREM, Lyon, France). Extrusion conditions were those already described by Nocente et al. [12]. For drying, pasta was arranged on frames and dried for 18 h, applying the conditions reported by [12]. The moisture of dried pasta was 12.5%.

### 2.3. Quality of the Cooked Pasta

Pasta samples were cooked according to the AACC method 16–50 [30]. The optimum cooking time (OCT) of pasta was determined when the white central core of the pasta just disappeared when squeezed between two test glasses, according to D’Egidio et al. [31]. Total organic matter (TOM) of pasta was determined according to D’Egidio et al. [32]. TOM values > 2.1 g/100 g correspond to low quality pasta, between 2.1 and 1.4 g/100 g correspond to good quality pasta, and <1.4 g/100 g correspond to very good quality pasta [33]. Water absorption (WA) was calculated from the weight increase of pasta at the OCT and determined as: WA = ((w − w_0_)/w_0_) × 100, where w and w_0_ were the weights of cooked and raw pasta, respectively [30]. Cooking loss (CL), expressed as grams of matter loss/100 g of raw pasta, was evaluated by weighing the residues of solids lost into the cooking water, after drying overnight at 105 °C. The residue was weighed and reported as percentage [30].

### 2.4. Sensory Testing

Sensory evaluation was performed, according to D’Egidio et al. [31], by a panel of three trained assessors who evaluated two textural characteristics: stickiness (material adhering to the cooked pasta surface) and firmness (resistance to chewing by the teeth). Each descriptor was scored from 10 to 100. The overall judgment (SJ) was calculated as the arithmetic mean of the scores of each descriptor.

### 2.5. Basic Composition and Total Antioxidant Capacity of BSG and Pasta Samples

All results are expressed as dry weight (dw) and the moisture content was determined using the thermo balance (Sartorius MA 40, Goettingen, Germany) at 120 °C. All analytical determinations were made in triplicate.

Protein contents of BSG and of BSG-enriched pasta samples were measured by micro-Kjeldhal nitrogen analysis according to ICC 105/2 method [34]. Total dietary fiber (TDF) content was determined using an enzymatic kit for fiber determination (Bioquant, Merck, Darmstadt, Germany) according to the Official Method 991.42 [35]. Ash content was determined according to approved method AACC 08-01.01 [36]. β-glucan content was evaluated by the Megazyme (Bray, Ireland) Mixed-Linkage Beta-Glucan kit [37].

Total antioxidant capacity (TAC) was determined by the “direct method,” according to Martini et al. [38].

### 2.6. Statistical Analysis

Results were expressed as mean ± standard deviation. One-way analysis of variance was performed with MSTATC program (Michigan State University, East Lansing, MI, USA), followed by the Duncan multiple range test for post-hoc comparison of means, applied to assess significant differences (*p* < 0.05) for each considered parameter.

## 3. Results and Discussion

### 3.1. Compositional Analysis of Raw Materials

Both spent grains from einkorn (BSGE) and tritordeum (BSGT) showed very high protein contents (Table 1), BSGE having a more than two-fold higher value than the value previously reported for barley BSG [5,12,39]. Likewise, BSGT had higher protein content than the mean value reported by Lynch et al. [5] (20%) for different sources of barley BSG. After all, also einkorn and tritordeum grains showed higher protein contents than barley [27,40]. On the contrary, both BSGE and BSGT’s total dietary fiber contents (Table 1) were lower than that reported in previous studies for barley spent grain. This result was expected since BSG from einkorn and tritordeum are devoid of the kernel husk, which contributes most to the higher fiber content recorded in barley BSG [5,12,39]. The β-glucan contents in BSGE and BSGT (Table 1) were lower than that found in barley BSG (2.18%) [12]; nevertheless, it was significantly higher than that registered in durum semolina (0.46%); hence, the addition of einkorn and tritordeum spent grain is supposed to improve the nutritional composition of enriched pasta. The higher values of β-glucan found in BSGT with respect to BSGE (Table 1) reflect the tritordeum genetic origin (barley × wheat). The large amounts of ash in BSG from einkorn and tritordeum (Table 1) represent the greater mineral contents of these cereal grains with respect to wheat and barley kernels [24,41]. The levels of TAC were significantly higher (96%) in BSGE and in BSGT (48%) than in durum semolina (Table 1). TAC values in einkorn were always higher than those found in durum wheat [25], likely due to the synergistic effects of antioxidant compounds such as tocols and carotenoids occurring in higher amounts in *T. monococcum* [23].

### 3.2. Chemical, Technological and Basic Characterizations of Dry Pasta

Though the addition of BSG rich in proteins, fibers and antioxidant compounds to durum semolina is assumed to improve the nutritional potential value of pasta, the percentage of replacement in bakery products is heavily limited by the negative effects of spent grain on the final quality of enriched products. Nevertheless, from our previous findings [12], the incorporation up to 10% of micronized barley BSG resulted in the best compromise in terms of technological, nutritional and sensorial aspects of enriched spaghetti. Then, in the present work, additions of 5 and 10 g of both micronized BSGE and BSGT to 100 g of durum wheat semolina were evaluated in terms of improvement of the nutritional composition of enriched pasta.

The additions of 5 and 10% BSG from einkorn and tritordeum to semolina increased the protein content by 1% on average, with pasta having 10% BSGE showing the highest value (Table 2). The ash content was similar in all pasta samples, even if a gradual increment was observed upon the addition of BSG (Table 2)—it stayed lower than 1.1% though.

The results showed that TDF content increased according to the magnitude of BSG enrichment. In particular, the amount of TDF was increased by 18% in pasta enriched with 5% BSG from einkorn or tritordeum and by 42% in BSGT10, and even by 68% in BSGE10 (Table 2), with respect to semolina pasta. Noteworthily, accordingly to Regulation (EC) number 1924/2006 [42], BSGE10 pasta could be referred as “rich in fiber,” being that it has 6 g dietary fiber/100 g product—the threshold for this nutritional and health claim; however, all pasta samples developed in this study could be labelled as “sources of fiber” due to their having higher than 3 g fiber/100 g. Only 10% BSG resulted in an increase in TAC value higher than the 10% in enriched pasta; lower increments, even if significant, were observed when 5% of semolina was replaced with BSG (Table 2). The addition of BSG to semolina resulted in the increasing of β-glucan content in enriched pasta. In particular, BSGE10 turned out to be increased by 52% with respect to the pasta control, whereas for BSGT10, the result was 74% (Table 2). Additionally, with 5% BSG, we registered an increment of β-glucan content in durum wheat pasta, the increment being more marked when BSG from tritordeum was added (22%) with respect to the addition of BSG from einkorn (9%). According to the health claim by the EFSA [43] related to β-glucan daily intake and its blood LDL cholesterol-lowering effect, the consumption of enriched BSG pasta could contribute to reaching the suggested optimal daily intake (3 g/die) of this bioactive compound.

### 3.3. Characterization and Sensory Evaluation of Cooked Pasta

Enrichments with 5 and 10% of both tritordeum and einkorn BSG had small but significant effects on TOM values (Table 3), which are associated, in each pasta sample, with very good quality, being lower than 1.4% [33]. Likely, the relative low fiber content brought by BSG did not cause a higher amount of starch to be released when cooking with respect to semolina pasta, since, at low concentrations, fiber might be dispersed into the protein/starch matrix. As confirmation, in our previous study [12], spaghetti enriched with different percentages of barley BSG turned out to be richer in fiber content (+20% on average, with respect to present formulations) and consequently resulted in a gradual increments of TOM values with more BSG. The control semolina pasta showed a lower cooking loss (Table 3) than enriched pasta, and statistically significant differences (*p* < 0.05) were observed amongst all samples, with BSGT5 and BSGT10 showing the highest values. Cooking loss parameter is one of the most important traits that affect consumer acceptance of fiber-enriched pasta. The increase in cooking loss in fiber-richer pasta is presumably due to weakening of protein network by the presence of TDF. Anyway, BSG-enriched pasta showed cooking losses below the values reported for good quality durum wheat pasta (<6.5%) [44].

A significant increase in optimal cooking time (OCT), determined by the disappearance of the starchy core, was observed upon the addition of BSG (Table 3), mainly in samples enriched with einkorn spent grain. An increase in cooking time also caused an increase in water absorption (Table 3), since more water can diffuse within starch and gluten, facilitating starch granules swelling and brokage [45].

The highest global sensorial judgment (Table 4) was found for durum semolina pasta and for pasta with 10% einkorn spent grain substitution; slightly lower scores, in the same class of quality, were reported for BSGT5 and BSGE5 (Table 4). Pasta with 10% BSGT replacement received the lowest scores in terms of both stickiness and firmness. It should be taken into account that the detrimental or positive effects of fiber addition to semolina depend also on the source of the fiber itself, as previously observed [45,46,47]. In particular, it could be inferred that the addition of einkorn spent grain affected to a minor extent the global quality of enriched pasta, likely due to higher protein content and lower β-glucan content than are found in tritordeum spent grains. Indeed, the inclusion of β-glucan in pasta formulation decreased the firmness and increased the stickiness value [48,49,50].

## 4. Conclusions

Many recent works have investigated the effects of adding spent barley grains to different cereal products. Currently, the growing interest in cereal alternatives to barley for the malting and brewing processes has made available considerable amounts of brewers’ spent grain of different plant origins. The present study indicated that the use of BSG, aimed at increasing the nutritional potential of pasta, should take into consideration the cereal species used in the malting and brewing processes. Indeed, pasta enriched with 10% spent einkorn grain showed the best quality in terms of technological performance, and the nutritional and sensorial parameters investigated. Nevertheless, the findings pointed out that the addition of micronized BSG from both cereals to semolina resulted in pasta with notable increases in protein, TDF and β-glucan content and to a minor extent in TAC levels, along with good sensorial quality. The increasing demand for healthier foods is encouraging the quest for novel raw materials and products. Further exploration should be aimed at considering the consumers’ attitude, in terms of flavor and mouthfeel properties, towards these new sorts of functional pasta, such as that proposed in the present paper. Moreover, it would be worth performing a cost–benefit economic analysis of the upcycling of BSG in the functional pasta formulations.

## Figures and Tables

**Figure 1 foods-10-00502-f001:**
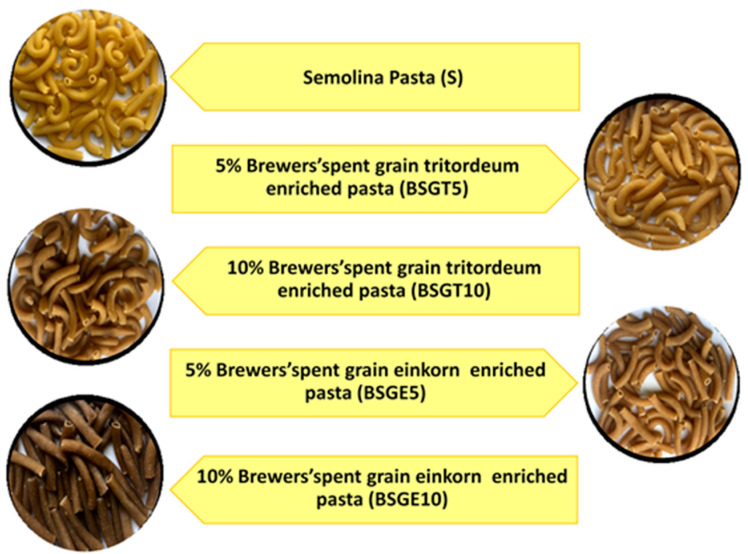
Dry pasta formulations: S = semolina 100%; BSGT5 = semolina:tritordeum Brewers’ spent grain (BSGT) 95:5 (*w*:*w*); BSGT10 = semolina:BSGT 90:10 (*w*:*w*); BSGE5 = semolina:einkorn BSG 95:5 (*w*:*w*); BSGE10 = semolina:BSGE 90:10 (*w*:*w*).

**Table 1 foods-10-00502-t001:** Basic composition of brewer spent grain (BSG) from einkorn and tritordeum.

SAMPLES	Protein	TDF	β-Glucan	Ash	TAC
	%	%	%	%	mmol TEAC/kg
**Einkorn BSGE**	32.5 ± 0.1	30.5 ± 0.3	1.00 ± 0.01	3.07 ± 0.02	60.8 ± 0.2
**Tritordeum BSGT**	21.6 ± 0.3	25.9 ± 0.3	1.660 ± 0.003	2.85 ± 0.02	45.8 ± 0.5

Results are reported as dry weight and expressed as the mean values ± standard deviations for three replications. TDF = total dietary fiber; TAC = total antioxidant capacity; TEAC = trolox equivalent antioxidant capacity.

**Table 2 foods-10-00502-t002:** Basiccomposition of control pasta (S) and brewers’ spent grain (BSG) enriched pasta (BSGE5, BSGE10, BSGT5 and BSGT10).

SAMPLES	Protein	Ash	TDF	TAC	β-Glucan
	%	%	%	mmol TEAC/kg	%
**S**	13.2 ± 0.4 c	0.866 ± 0.008 e	3.4 ± 0.1 d	31 ± 2 c	0.46 ± 0.07 d
**BSGE5**	14.3 ± 0.2 b	0.963 ± 0.005 c	4.0 ± 0.3 c	34.0 ± 0.7 b	0.50 ± 0.03 cd
**BSGE10**	15.2 ± 0.3 a	1.098 ± 0.002 a	5.7 ± 0.4 a	37 ± 1 a	0.702 ± 0.001 b
**BSGT5**	13.8 ± 0.2 bc	0.909 ± 0.008 d	4. 0 ± 0.4 c	33 ± 2 bc	0.56 ± 0.04 c
**BSGT 10**	14.3 ± 0.4 b	1.000 ± 0.005 b	4.8 ± 0.4 b	34.6 ± 0.6 b	0.81 ± 0.05 a

Results are reported as dry weight and expressed as mean ± standard deviation for 3 replications. Within the same column, values with different letters indicate significant differences determined by Duncan’s test (*p* < 0.05). S: semolina 100%; BSGE5: semolina/BSG einkorn (95:5); BSGE10: semolina/BSG einkorn (90:10); BSGT5: semolina/BSG tritordeum (95:5); BSGT10: semolina/BSG tritordeum (90:10). TDF = total dietary fiber; TAC = total antioxidant capacity; TEAC = trolox equivalent antioxidant capacity.

**Table 3 foods-10-00502-t003:** Cooking properties of pasta with different blends of BSG/semolina.

SAMPLES	TOM	CL	OCT	WA
	%	%	min′ sec″	g
**S**	1.28 ± 0.08 a	4.61 ± 0.04 e	7′30″ ± 5″ c	181.80 ± 0.02 e
**BSGE5**	1.07 ± 0.04 bc	4.78 ± 0.09 d	8′10′′ ± 5″ a	185.80 ± 0.01 c
**BSGE10**	1.00 ± 0.04 c	4.85 ± 0.01 c	8′10′′ ± 5″ a	197.00 ± 0.03 a
**BSGT5**	1.01 ± 0.09 bc	5.14 ± 0.04 a	7′50′′ ± 5″ b	184.20 ± 0.02 d
**BSGT10**	1.10 ± 0.01 b	4.95 ± 0.02 b	8′00′′ ± 5″ ab	193.50 ± 0.01 b

Results are reported as dry weight and expressed as mean value ± standard deviation for 3 replications. Within the same column, values with different letters indicate significant differences determined by Duncan’s test (*p* < 0.05). S: semolina 100%; BSGE5: semolina/BSG einkorn (95:5); BSGE10: semolina/BSG einkorn (90:10); BSGT5: semolina/BSG tritordeum (95:5); BSGT10: semolina/BSG tritordeum (90:10). TOM: total organic matter; CL: cooking loss; OCT: optimal cooking time; WA: water absorption.

**Table 4 foods-10-00502-t004:** Sensory evaluation of pasta samples.

SAMPLES	Cooking Quality Parameters		
	Stickiness	Firmness	Global Sensorial Judgement
**S**	90	75	83
**BSGE5**	80	75	78
**BSGE10**	85	80	83
**BSGT5**	85	70	78
**BSGT10**	70	70	70

S: semolina 100% (control); BSGE5: semolina-BSG einkorn (95:5); BSGE10: semolina-BSG einkorn (90:10); BSGT5: semolina-BSG tritordeum (95:5); BSGT10: semolina-BSG tritordeum (90:10). For stickiness: 10–20 = very high, 21–40 = high, 41–60 = rare, 61–80 = minimal and 81–100 = absent; for firmness: 10–20 = absent, 21–40 = rare, 41–60 = sufficient, 61–80 = good and 81–100 = very good. For global sensorial judgment, 54 = scarce; 55–64 = sufficient; 65–74 = good; 75–100 = very good.

## Data Availability

Data is contained within the article.

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
