# Peer review of "Using Einkorn and Tritordeum Brewers’ Spent Grain to Increase the Nutritional Potential of Durum Wheat Pasta"

_foods, 2021, doi:10.3390/foods10030502_

Round 1

Reviewer 1 Report

  1. Cooking terminology should be re-defined.
  2. Aroma, flavour, and good mouthfeel properties should meet consumers’ requests, Explain.
  3. Financial comparison between the different processes of producing normal pasta and that suggested, would be added.
  4. The authors should support and argument their notion of why to use basta after adding the brewers’ spent grain.
  5. Please follow the journal instructions of author.
  6. The coming reference may help in the introduction.

Wu, D., Tu, M., Wang, Z., Wu, C., Yu, C., Battino, M., El-Seedi, H.R. and Du, M., 2020. Biological and conventional food processing modifications on food proteins: Structure, functionality, and bioactivity. Biotechnology advances, 40, p.107491.

Author Response

Reviewer # 1

  1. Cooking terminology should be re-defined.

Answer: It is not very clear what the reviewer refers to with ‘cooking terminology’, since we used scientific food technology terminology. Anyway, we inserted Table 4 (line 264 of the revised MS) with the definition of sensorial parameters for cooked pasta.

  1. Aroma, flavour, and good mouthfeel properties should meet consumers’ requests, Explain.

Answer: We didn’t test  aroma, flavour and mouthfeel in the present paper but we considered only parameters related to technological quality of cooked pasta. However, a sentence was added (line 288-290 of the revised MS)  to better address possible future investigations about consumers attitude.

  1. Financial comparison between the different processes of producing normal pasta and that suggested, would be added.

Answer: We understand reviewer concern, but economic valuation is not one of the aims of the present study, neither our area of expertise. Nevertheless, we added a sentence (line 290-292 of the revised MS) to stress the importance of a costs/benefits analysis to introduce these novel pasta formulations in the market.

  1. The authors should support and argument their notion of why to use basta after adding the brewers’ spent grain.

Answer: We added a sentence at lines 78-80 of the revised MS

  1. Please follow the journal instructions of author.

Answer : we checked again and followed the author’s instructions throughout the MS

  1. The coming reference may help in the introduction.

Wu, D., Tu, M., Wang, Z., Wu, C., Yu, C., Battino, M., El-Seedi, H.R. and Du, M., 2020. Biological and conventional food processing modifications on food proteins: Structure, functionality, and bioactivity. Biotechnology advances, 40, p.107491.

Answer: In our work we didn’t deeply investigate protein modifications over pasta-making and cooking  process.

Reviewer 2 Report

Review of Manuscript Number: ID: Foods-1112148 peer rewiew v1

Title:Einkorn and tritordeum brewers’ spent grain to increase the nutritional potential of durum wheat pasta

Journal: Foods

The research conducted by the authors of the manuscript is interesting and important. They concern an important problem, which is the management of by-products of the food industry. One such product is brewers' spent grain (BSG). It's the major by-product of brewing industry. It's can be used as functional ingredient to increase the nutritional value of cereal based products. In this work, micronized BSG from einkorn and tritordeum brewing process were characterized and used to produce four macaroni pasta formulations enriched with BSG at ratios of 5 g and 10 g / 100 g of semolina. Einkorn BSG showed the highest values ​​for all the analyzed parameters, proteins, total dietary fiber (TDF) and Total Antioxidant Capacity (TAC), except for β-glucan.

The addition of BSG from einkorn and tritordeum affected to a minimal extent the sensory properties of cooked macaroni, which showed higher values ​​of optimal cooking time and cooking loss, but lower total organic matter compared to semolina pasta. The solution proposed by the authors allows to obtain pasta with potential health-promoting properties, while maintaining the organoleptic properties that are beneficial for the consumer.

In conclusion, I recommend the manuscript to be published. I suggest minor editorial changes:

line 11– Total Dietary Fiber should be total dietary fiber

line 12 – Total Antioxidant Capacity – should be- total antioxidant capacity

line 110 – Pasta cookin quality –should be – quality of the cooked pasta

line 119 - Provide a reference for the determination of water absorption (WA) and cooking loss (CL)

line 132 – proximate composition of BSG – should be “basic composition of BSG

line 143 - The TAC is a material property, not a component. A separate section 2.6 should be distinguished - it describes the method of determining the total oxidation potential.

line 174 - explain TEAC abbreviation under the table

line 177 - replace "proximete" with "basic"

Author Response

Reviewer #2

  1. The authors discussed an interesting topic in this manuscript. Nevertheless, in my opinion, too few results have been presented to maintain this type of work. I suggest publishing this work as 'Communication

Answer: We don’t agree. In the present article, results about several nutritional, technological and sensorial characteristics starting from a novel raw material to cooked pasta, were presented.

  1. The BSG processing methods are also important. Properly selected, they enable the acquisition of valuable nutrients (eg DOI: 10.3390/su12093660).

Answer: We added the reference suggested (line 41 and lines 312-313 of the revised MS)

  1. This is an interesting novelty. This cereal is not common, there is still ongoing breeding work to obtain new lines of this cereal. There is still little literature data on the properties of Tritordeum. In my opinion, it is worth describing here more broadly - in a few sentences - its nutritional and health-promoting properties

Answer: We added some information about the nutritional properties of tritordeum (lines 65-71 of the revised MS)

  1. please use one term - I suggest 'pasta'

Answer: We substitute the term macaroni with  pasta almost throughout the revised MS

  1. Are these materials from own beer production or obtained from a commercial brewery?

Answer: we added a sentence (line 84  of the revised MS) to better specify the brewing process performed to obtain the BSG

  1. to what value?

Answer:  It specified at lines 99-100 of the revised MS

  1. Please note the number of significant decimal places. I recommend that you use the same number consistently throughout the table.

Answer: In number quotation we follow the rules of Taylor 1997. Error analysis: The study of uncertainties in physical measurements. Sausalito, CA: University Science Books. In particular: we applied the following rules: 1. The mean cannot be more accurate than the original measurement 2. The mean has the same significant digits as standard deviation  which determines the number of significant digits 3. Standard deviation has been rounded to one significant digit (first value different to zero).

  1. I believe this data should be removed from the article. Even enriched pasta does not have high antioxidant potential. In addition, the most important thing would be the activity of the cooked pasta, as thermal treatment may also significantly affect the obtained values. Either positively or negatively.

Answer: we do not agree about the omission of TAC results. Indeed, the paper deals with the characterization of a very novel raw material (Einkorn and tritordeum brewer spent grains), to the best of our knowledge, never performed before, hence also the TAC values data could be interesting for the scientific community.  You are correct that we determined TAC values only in dry pasta, indeed  in cooked pasta we determined just the cooking quality; anyway, in a future work we are planning to characterize better cooked pasta in terms of stability of bioactive compounds. Moreover, in our previous work (Martini et al. 2018 International J of food Science and Nutrition , 69, 24-32) we observed that in whole meal pasta, where the outmost kernel layers were present, the TAC values improved over cooking and we can infer that the same could happen in present products where the external layers of the kernel are  present,too.

  1. die?

Answer: Die stands for day in the medical dictionary speaking about posology and dosage

  1. I have doubts about the statistical analysis. Please recalculate everything and verify that it was calculated correctly.

Answer: You are correct. We verified and found some wrong digits. We inserted the correct analysis in the revised MS.

  1. Where is this table and those results?

Answer : In the revised MS we inserted Table 4 (lines: 264-270)

  1. Need to explain how physicochemical properties (presented in this work) affected consumer acceptance. Relationship among them may be revealed by, for example, a PCA biplot.

Answer: Yes we agree, but we didn’t perform consumers acceptance in this work. We took into account the judgment of stickiness and firmness parameters made by the three trained assessors, who are technicians and not consumers, then we have  very few sensorial parameters to perform a PCA.

  1. I will not agree that it was a very big increase. Please omit the TAC results throughout the manuscript.

Answer: Yes, for sure it was not a big increase, but as explained in the previous answer (n.8), it is a data; however, we changed the sentence at line 285-286 of the revised MS.

Reviewer 3 Report

Comments are included in the file.

Author Response

Reviewer #3

  1. line 11– Total Dietary Fiber should be total dietary fiber

Answer: we corrected at Line 11-12 of the revised MS

  1. line 12 – Total Antioxidant Capacity – should be- total antioxidant capacity

Answer: we corrected at Line 11-12 of the revised MS

  1. line 110 – Pasta cookin quality –should be – quality of the cooked pasta

Answer : we corrected at Line 114 of the revised MS

  1. line 119 - Provide a reference for the determination of water absorption (WA) and cooking loss (CL)

Answer: we added the references for the two determination . Lines 125 128 of the revised MS

  1. line 132 – proximate composition of BSG – should be “basic composition of BSG

Answer: We changed ‘proximate’ with ‘basic’ at line Line 136 of the revised MS

  1. line 143 - The TAC is a material property, not a component. A separate section 2.6 should be distinguished - it describes the method of determining the total oxidation potential.

Answer: we changed the title of the paragraph to follow your suggestion, at line Line 136 of the revised MS

  1. line 174 - explain TEAC abbreviation under the table

Answer : Done, we explain TEAC abbreviation below the Tables

  1. line 177 - replace "proximate" with "basic"

Answer: we replaced ‘proximate’ with ‘basic’ at line 214 (Table 2)

Round 2

Reviewer 3 Report

The authors responded correctly to most of my comments. Nevertheless, they disagreed with a few. I am glad that they are able to defend their opinion and formulated their answers convincingly.

I have no more comments.